

# Identification of 4-aminoquinoline core for the design of new cholinesterase inhibitors

Yao Chen[1,2], Yaoyao Bian[3], Yuan Sun[4], Chen Kang[5], Sheng Yu[1], Tingming Fu[1,2], Wei Li[1], Yuqiong Pei[1] and Haopeng Sun[6]

[1] School of Pharmacy, Nanjing University of Chinese Medicine, Nanjing, China
[2] Jiangsu Collaborative Innovation Center of Chinese Medicinal Resources Industrialization, Nanjing University of Chinese Medicine, Nanjing, China
[3] School of Nursing, Nanjing University of Chinese Medicine, Nanjing, China
[4] Department of Chemistry and Biochemistry, Ohio State University, Columbus, OH, United States
[5] Division of Pharmacology, College of Pharmacy, Ohio State University, Columbus, OH, United States
[6] Department of Medicinal Chemistry, China Pharmaceutical University, Nanjing, China

## ABSTRACT

Inhibition of acetylcholinesterase (AChE) using small molecules is still one of the most successful therapeutic strategies in the treatment of Alzheimer's disease (AD). Previously we reported compound T5369186 with a core of quinolone as a new cholinesterase inhibitor. In the present study, in order to identify new cores for the designing of AChE inhibitors, we screened different derivatives of this core with the aim to identify the best core as the starting point for further optimization. Based on the results, we confirmed that only 4-aminoquinoline (compound 04 and 07) had cholinesterase inhibitory effects. Considering the simple structure and high inhibitory potency against AChE, 4-aminoquinoline provides a good starting core for further designing novel multifunctional AChEIs.

## INTRODUCTION

Alzheimer's disease (AD), an age-related and progressive neurological disease, severely threatens the health of elderly human beings (*Palmer, 2011*). It leads to impairment in memory, language skills, judgment and orientation (*Goedert & Spillantini, 2006*), and accounts for nearly 70% of adult dementia (*Castellani, Rolston & Smith, 2010*). Worst of all, it severely burdens the social health service considering that the prevalence of AD will rise significantly in the next several decades (*Reitz, Brayne & Mayeux, 2011*). So far, the etiology of AD is not fully understood, but several common hallmarks, including cholinergic dysfunction (*Scarpini, Scheltens & Feldman, 2003*), amyloid-$\beta$ (A$\beta$) deposits (*Terry, Gonatas & Weiss, 1964*), $\tau$-protein aggregation (*Grundke-Iqbal et al., 1986*), oxidative stress (*Wilson et al., 2013*), neuroinflammation (*Linker et al., 2011*), excitotoxicity (*Kaidery et al., 2013*), calcium impairment (*Diaz et al., 2009*), mitochondrial dysfunction (*Aliev et al., 2014*), have been reported to tightly correlated to the development of AD. These findings provide researchers multiple choices to design treating agents for AD.

Corresponding authors
Yuqiong Pei, peiyuqiong@126.com
Haopeng Sun, sunhaopeng@163.com

Although many mechanisms as well as active compounds have been reported, only two classes of drugs, acetylcholinesterase inhibitor (AChEI) and N-methyl-D-aspartate receptor (NMDAR), are clinially available for AD treatment. The enzymatic cavity of AChE has the shape of a nearly 20 Å deep narrow groove which is composed of two binding sites. The one is catalytic active site (CAS) at the bottom of the binding pocket. It mediates the hydrolysis of acetylcholine (*Muñoz Ruiz et al.*, *2005*). The other is the peripheral anionic site (PAS) near the entrance of the gorge. PAS has been considered to have close relation to both hydrolysis of acetylcholine and neurotoxic cascade of AD through AChE-induced $\beta$-amyloid (A$\beta$) aggregation (*Terry, Gonatas & Weiss*, *1964*). Recently, it is widely accepted that multi-functional AChEIs, also known as "multi-target-directed ligands" (MTDLs), have advantages to enhance the inhibitory potency of AChEIs. MTDLs means compounds with additional properties other than cholinesterase inhibition through targeting different drug targets. They are recognized as promising agents for AD treatment (*Muñoz Ruiz et al.*, *2005*). However, in order to modulate different targets simultaneously, MTDLs need multiple pharmacophoric features, leading to their structures very complicated, and many of them exhibit high molecular weight and LogP, which may cause potential problems in further development. Therefore, acquiring simple and potent structures with high ligand efficiency (LE) as starting point to design MTDLs against AChE is an attractive task for medicinal chemists.

Previously, we have reported compound **T5369186** (**23**) as a new cholinesterase inhibitor from shape-based virtual screening with tacrine as template (*Chen et al.*, *2015*). The compound contains a simplified quinoline core compared to tacrine. Considering that quinoline is a privilege core in drug molecules, especially showing activity against cancer, infective and degenerative diseases (*Solomon & Lee*, *2011*; *Graves et al.*, *2002*), we think this core provides us a good starting point for the identification of new AChE inhibitors. To give a further structural analysis of this core on the inhibition of AChE, herein we describe our efforts to further confirm the pharmacophoric determinants of this core.

## EXPERIMENTAL METHODS

### *In vitro* cholinesterase Inhibition Assay

The assay followed the method of *Ellman et al.* (*1961*) using a Thermo Scientific Varioskan Flash. AChE (C3389, Type VI-S, from Sigma) and BuChE (C0663, from human erythrocytes), 5,5′-dithiobis (2-nitrobenzoic acid) (Sigma reagent, DTNB, D218200), acetylthiocholine (ATC), and butyrylthiocholine (BTC) iodides were purchased from Sigma-Aldrich (Shanghai, China). AChE/BuChE stock solution was prepared by adjusting 500 units of the enzyme and 1 mL of gelatin solution (1% in water) to 100 mL with water. This enzyme solution was further diluted before use to give 2.5 units/mL. ATC/BTC iodide solution (0.075 M) was prepared in water. DTNB solution (0.01 M) was prepared in water containing 0.15% ($w/v$) sodium bicarbonate. For buffer preparation, potassium dihydrogen phosphate (1.36 g, 10 mmol) was dissolved in 100 mL of water and adjusted with KOH to pH = 8.0 ± 0.1. Stock solutions of the test compounds were prepared in ethanol, 100 μL of which gave a final concentration of $10^{-4}$ M when diluted to the

final volume of 132 μL. For each compound, a dilution series of at least five different concentrations (normally $10^{-4} \sim 10^{-9}$ M) was prepared.

For measurement, a cuvette containing 100 μL of phosphate buffer, 10 μL of the respective enzyme, and 10 μL of the test compound solution was allowed to stand for 5 min before 10 μL of DTNB were added. The reaction was started by addition of 2 μL of the substrate solution (ATC/BTC). The solution was mixed immediately, and exactly 2 min after substrate addition the absorption was measured at 25 °C at 412 nm. For the reference value, 10 μL of water replaced the test compound solution. For determining the blank value, additionally 10 μL of water replaced the enzyme solution. Each concentration was measured in triplicate at 25 °C. The inhibition curve was obtained by plotting percentage enzyme activity (100% for the reference) versus logarithm of test compound concentration. Calculation of the $IC_{50}$ values was performed with Graph Pad Prism 5.0.

## Kinetic study

Kinetic measurements were performed in the same manner, while the substrate (ATC/BTC) was used in concentrations of 25, 50, 90, 150, 226, 452 and 678 μM for each test compound concentration and the reaction was extended to 4 min before measurement of the absorption. Vmax and Km values of the Michaelis–Menten kinetics were calculated by nonlinear regression from substrate-velocity curves using Graph Pad Prism 5.0. Linear regression was used for calculating the Lineweaver–Burk plots.

## Molecular docking

Computational methods are useful tools for drug discovery and evaluation that has been widely applied in drug discovery campaign (*Zheng et al.*, *2013*; *Ford & Ho*, *2016*). The docking study was performed by CDOCKER module implemented in Discovery Studio 3.0. The principle of CDOCKER can be breifly summarized as follow: CDOCKER generates ligand "seeds" to populate the binding pocket. Each seed is then subjected to high temperature molecular dynamics (MD) using a modified version of CHARMm force field (*Wu et al.*, *2003*). The structure after MD run is then fully minimized under the forcefield. The solutions are then clustered according to position and conformation and ranked by energy. The cocrystal structure of Torpedo Californica AChE bound with bis(7)-tacrine (TcAChE, PDB id: 2CKM) was used for molecular docking. The binding sites were defined by residues around the CAS and PAS of AChE (in 6 Å radius). The heating step, cooling steps, and cooling temperature were set to 5,000, 5,000, and 310, respectively. Other parameters were kept as default.

## Compound information

All compounds except **03**, **10**, **15**, **17** and **22** were purchased from Sigma Aldrich (Sigma Aldrich, Shanghai, China: http://www.sigmaaldrich.com/china-mainland.html), with purity >95.0%. Compounds **03**, **10**, **15**, **17** and **22** were bought from J&K Scientific (J&K Scientific, Shanghai, China: http://www.jkchemical.com/). The detailed information is listed in Table S1.

## RESULTS

### Identification of 4-aminoquinoline as the potent core targeting AChE

Multiple aminoquinolines, with amino group substituted at different position of quinolone ring (Table 1, compound **01**∼**07**), were firstly collected. Preliminary evaluations of these compound were performed by determine their AChE (eeAChE) % inhibition at 10 µM following Ellman's method (*Ellman et al.*, *1961*). It was previously reported that **T5369186** (**23**) and tacrine were used as positive controls. Results showed that only 4-aminoquinoline core had strong AChE inhibition (% inhibition of **03** and **07** $68.29 \pm 1.83\%$ and $90.59 \pm 0.28\%$, respectively). Compound **07** was further determined for the AChE inhibitory curve and $IC_{50}$ (**23** used as positive control). The compound showed dose-dependent manner and well fitted inhibitory curve (Fig. 1), with $IC_{50}$ $0.72 \pm 0.06$ µM.

### Kinetics study

To gain information on the mechanism of inhibition, compound **07** was selected for kinetic studies of AChE inhibition by using Lineweaver–Burk plots, which were reciprocal rates versus reciprocal substrate concentrations for the different inhibitor concentrations resulting from the substrate-velocity curves for AChE (Fig. 2). The compound exhibited a mixed-type inhibition of AChE, for the plot showed both increased slopes (decreased Vmax) and intercepts (higher Km) when the concentration of the inhibitors were increased, indicating that the compounds may bind to both CAS and PAS (*Chen et al.*, *2015*). The detailed Km and Vmax values from the non-linear regression fitting in Lineweaver–Burke is shown in Table S2.

### Structure–activity relationship and binding mode analysis by molecular docking

To deeply understand the binding mode of between AChE and the potent compounds, molecular docking was applied to further analyze compound **07** and **23** (Fig. 3). The binding conformation suggested that the two compounds bound to catalytic site (CAS) of AChE in a very similar manner. In detail, **07** formed strong $\pi$–$\pi$ interactions with Trp84, Phe330 and Tyr334 of the CAS of AChE. The amino group formed a H-bond with His440, which was considered as a critical member of the catalytic triad of AChE. The methyl group inserted into a small sub-pocket surrounded by Asp72 and Ser81 and contacted the backbone of them through hydrophobic interactions, which enhanced the activity of the inhibitor. Compound lacked this group (**03**, $68.29 \pm 1.83\%$ inhibition at 10 µM) showed reduced activity, and this further confirmed that methyl was a pharmocophoric group for 4-aminoquinoline core. The acetyl group of **23** inserted into a hydrophobic groove formed by the aromatic side chains of Trp84, Phe330 and Tyr334, which contributed to the binding affinity of **23** (region in the red dot line in Fig. 3). However, this group was missed in **07**, leaving the pocket unoccupied. This could be the reason for the decreased activity of compound **07** compared to **23**.

Ligand efficiency (LE) is an important parameter when evaluate the advantage of lead compounds or active fragments (*Reynolds, Tounge & Bembenek*, *2008*). It is an attempt

**Table 1** The preliminary assay of collected compounds with aminoquinoline or other similar cores.

| Cpd. | Structure | AChE % inhibition[a] |
|------|-----------|---------------------|
| 01 | | $4.88 \pm 1.46$ |
| 02 | | $18.89 \pm 2.77$ |
| 03 | | $68.29 \pm 1.83$ |
| 04 | | $13.14 \pm 3.48$ |
| 05 | | $4.02 \pm 2.96$ |
| 06 | | $3.21 \pm 2.32$ |
| 07 | | $90.59 \pm 0.28$ |
| 08 | | $16.04 \pm 2.72$ |
| 09 | | $13.55 \pm 2.0$ |

**Table 1** (*continued*)

| Cpd. | Structure | AChE % inhibition[a] |
|---|---|---|
| 10 | | 37.51 ± 1.52 |
| 11 | | 31.74 ± 2.17 |
| 12 | | 17.36 ± 1.98 |
| 13 | | 3.32 ± 1.79 |
| 14 | | 4.26 ± 4.43 |
| 15 | | 13.64 ± 3.29 |
| 16 | | 33.15 ± 3.87 |
| 17 | | 35.88 ± 4.01 |
| 18 | | 26.02 ± 6.82 |
| 19 | | 7.56 ± 2.11 |

**Table 1** (*continued*)

| Cpd. | Structure | AChE % inhibition[a] |
|------|-----------|----------------------|
| **20** | | $16.63 \pm 3.01$ |
| **21** | | $-8.59 \pm 1.23$ |
| **22** | | $5.48 \pm 0.77$ |
| **23** | | $98.53 \pm 1.81$ |
| **Tacrine** | | $97.36 \pm 2.54; 0.08 \pm 0.01^{b}$ |

**Notes.**
[a] % inhibition at 10 $\mu$M.
[b] $IC_{50}$ of tacrine ($\mu$M).

**Table 2** The $IC_{50}$, CLogP and ligand efficiency of active compounds.

| Cpd. | AChE $IC_{50}$($\mu$M) | CLogP[a] | LEMW[b] |
|------|----------------------|----------|---------|
| **07** | $0.72 \pm 0.06$ | 1.43 | 0.039 |
| **23** | $0.57 \pm 0.09$ | 1.64 | 0.032 |

**Notes.**
[a] ClogP is predicted by MarvinSketch 5.10.0 with all the parameter set as default.
[b] LEMW stands for ligand efficiency based on molecular weight, LEMW = $-pIC_{50}$/MW.

to normalize the activity of a compound by its molecular size (*Congreve et al.*, *2008*). To further recognize the importance of the acetyl group, ligand efficiency was calculated for **07** and **23** based on their $-pIC_{50}$ and molecular weight (Table 2). Although **23** was slightly more potent than **07**, the LE of **23** was lower than **07**, indicating that the acetyl group was useful but not important to the inhibitory activity. Considering the hydrophobic character of the AChE sub-pocket around this group, proper optimization, especially those groups easily to form hydrophobic contacts, may help to further enhance the activity as well as LE. The CLogP value of **07** and **23** was also predicted (1.43 and 1.64, respectively). Considering that the further design of MTDLs based on the core will enhance the CLogP because of the introduction of hydrophobic groups, the initial CLogP value of the two compounds is acceptable.

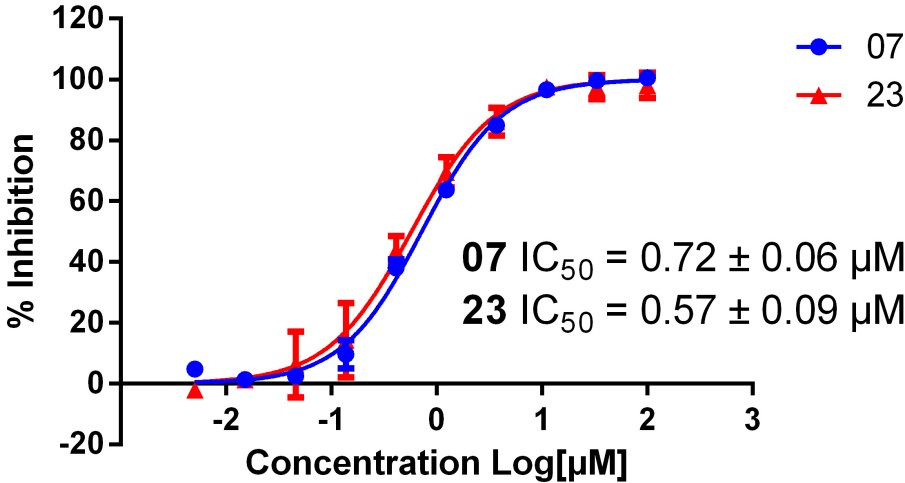

**Figure 1** **Inhibitory curve of compound 07 and 23 on AChE.** Calculated $IC_{50}$s of the two compounds are shown. The initial concentration was set as 100 $\mu$M and then at 5 times dilution for another nine concentrations.

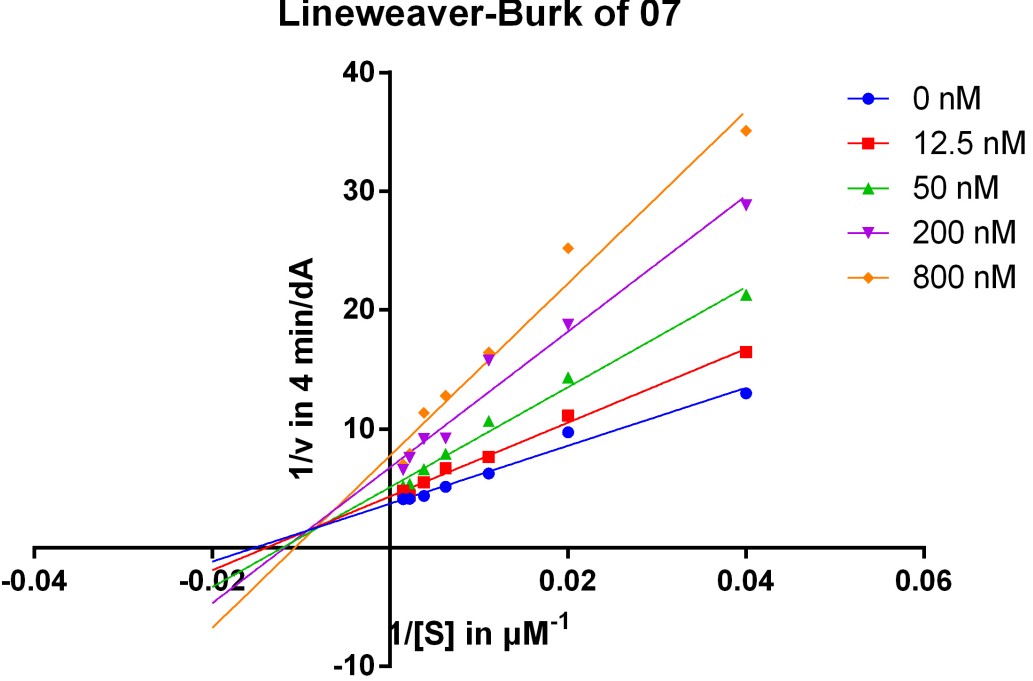

**Figure 2** Lineweaver-Burk plots of compound **07** resulting from subvelocity curves of AChE activity with different substrate concentrations (25 $\sim$ 678 $\mu$M) in the absence and presence of 12.5, 50, 200, 800 nM of the compound.

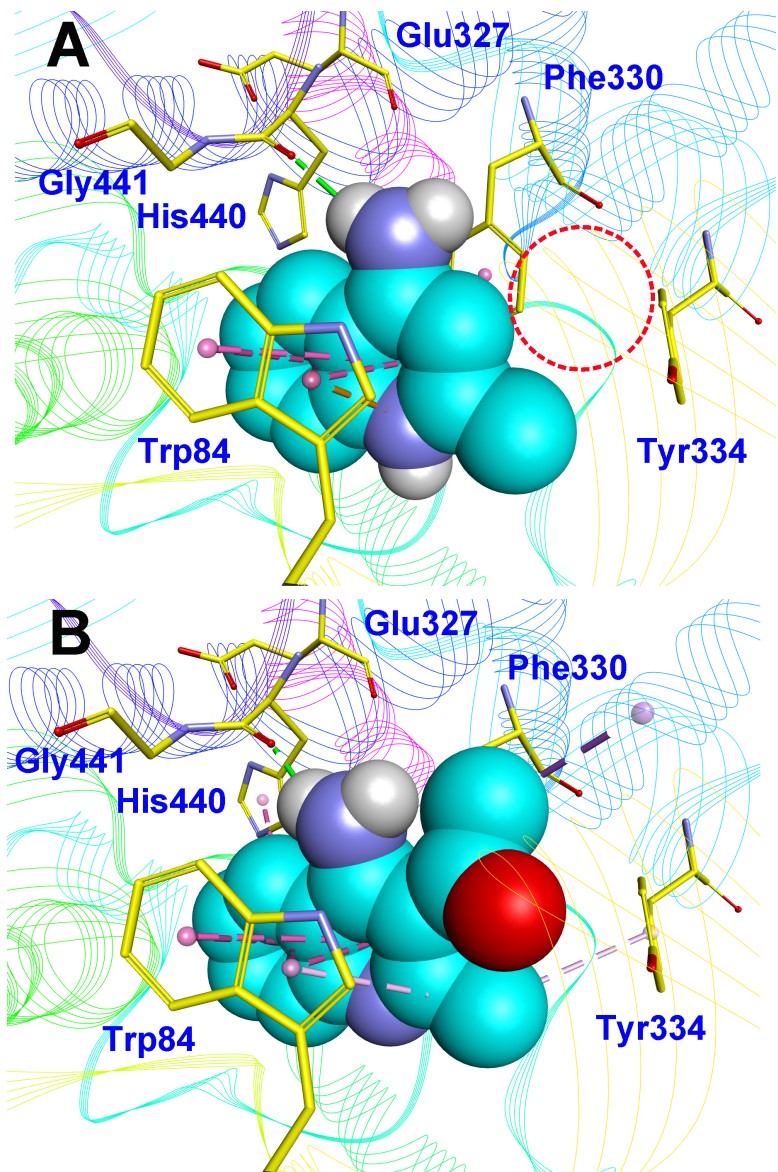

**Figure 3** **Binding mode prediction of 07 (A) and 23 (B) with AChE (PDB id: 2CKM).** Compounds were shown in blue CPK mode (carbon atoms), key residues were shown in yellow stick mode. Hydrophobic contact and $\pi$-$\pi$ stacking were depicted in purple dot line, H-bonds were in green dot line. Only polar hydrogens of the compounds were shown.

Amino group at other position of quinoline (**01, 02, 04 ~ 06**), however, exhibited remarkably reduced activity, with % inhibition ranging from $3.21 \pm 2.32$ to $33.51 \pm 1.52\%$ at 10 μM. The results indicated that spatial location of the amino group on quinolone ring was one of the determinants to the activity of the compound. According to the previously reported results, the catalytic triad including Ser200, Glu327 and His440 played a critical role in hydrolyzing acetylcholine by AChE. Inhibitors directly interacting or closing to this triad can impede the catalytic function of AChE. According to the binding mode (Fig. 3), the amino group at 4-position of quinolone pointed to Glu327, and interacted with His440.

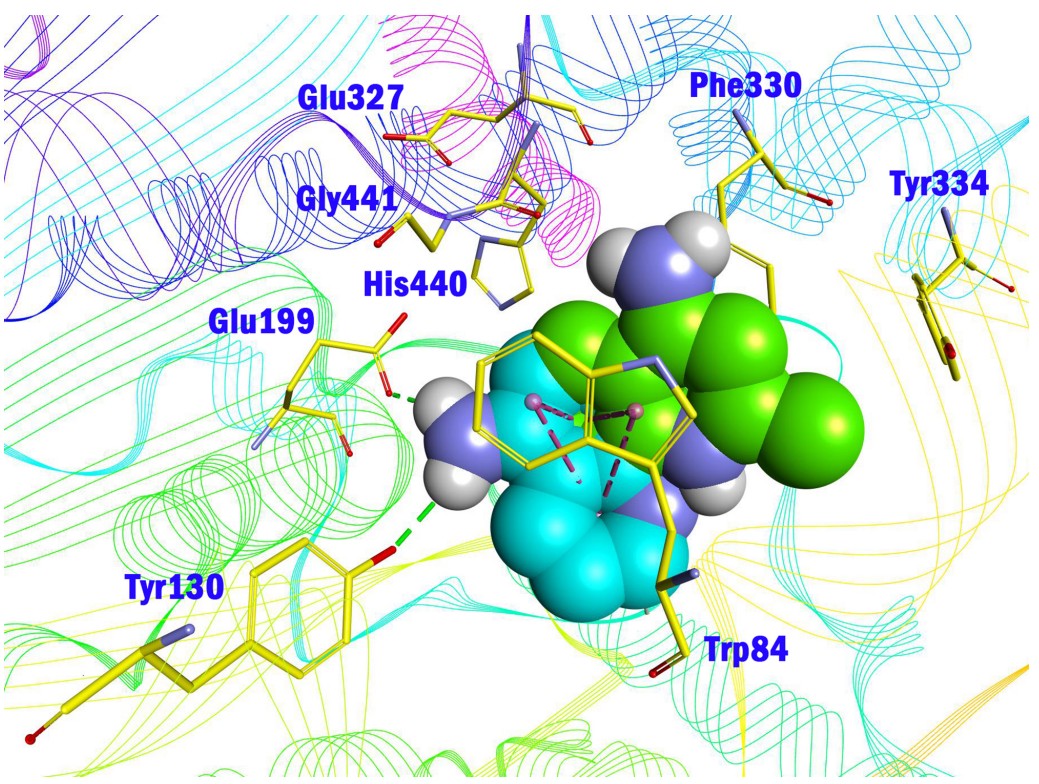

**Figure 4** **Comparison of the binding mode of 04 and 07 in the CAS of AChE (PDB id: 2CKM).** Compounds **04** and **07** were shown in CPK mode and colored by blue and green (carbon atoms), respectively. Key residues were shown in yellow line mode. Hydrophobic contact and $\pi$-$\pi$ stacking were depicted in purple dot line, H-bonds were in green dot line.

This mode can inhibit the approach of acetylcholine to the catalytic site, thus exerted AChE inhibition. Oppositely, **04** (Fig. 4, blue) bound to AChE with a completely different mode from **07** (Fig. 4, green). The 5-amino group pointed to Glu199 and Tyr130 and formed two H-bonds to the residues, leading to a different location of the quinolone ring compared to that of compound **07**. Although it interacted with Trp84 through $\pi-\pi$ stacking, it was far from Phe330 and Tyr334, and did not form any interaction with these residues, which were important when **07** bound to AChE. Under such binding mode, **04** was moved away from the catalytic triad, and could not inhibit the approach of acetylcholine to the CAS of AChE; therefore, **04** exhibited very poor inhibitory activity. This can also be the reason for the loss of the activity of other aminoquinolines. Additionally, the results indicated that the CAS site was large enough to endure structural modification of 4-aminoquinoline ring, especially on the benzene ring. Proper optimization at this site could improve the binding affinity of the compound through forming polar recognitions or hydrophobic contacts towards the sub-pocket around Tyr130 and Glu199.

To further confirm the importance of 4-amino group, we replaced it with other substituents including halogen, hydroxyl, carboxyl and nitro groups (**08**~**12**). Only 4-hydroxyl and 4-carboxyl exhibited moderate inhibition, while 4-chloro, 4-bromo and
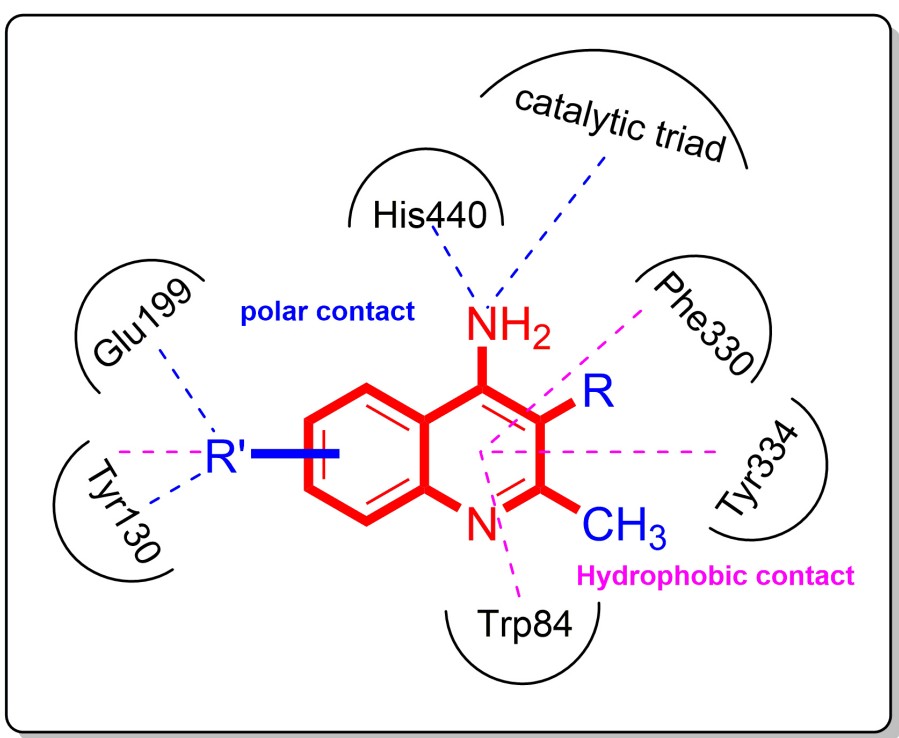

**Figure 5   Structural determinants and modification strategy of 4-aminoquinoline core.** (Red color stands for structural determinants, while blue color stands for groups that can be optimized).

4-nitro compounds loss the inhibitory activity. The results further confirmed that polar contact near the catalytic triad of CAS was a determinant for AChE inhibition.

Next we changed the core to other rings including isoquinoline (**13** ~ **15**), naphthalene (**16** ~ **17**), 5,6,7,8-tetrahydronaphthalene (**18**) and 1*H*-indole (**19** ~ **22**) to verify the function of quinolone ring. The isoquinoline and 1*H*-indole, which had different arrangement of the polar group or atom to that of quinolone, led to completely loss of inhibitory activity, indicating that electrostatic character of the bicyclic core was another determinant for AChE inhibition.

## DISCUSSION AND CONCLUSIONS

In conclusion, we identified 4-aminoquinoline as the basic core for the design of new cholinesterase inhibitors. Structural determinants and modification strategy (summarized in Fig. 5) were discussed in this article. Small hydrophobic groups at 2- and 3- position of quinolone improve the binding affinity through hydrophobic contacts. Additionally, appropriate substituents at benzene ring (R′ position in Fig. 5) can be introduced to fit the sub-pocket around Tyr130 and Glu199. Considering the simple structure and high inhibitory potency against AChE, 4-aminoquinoline provides a good starting core for further designing novel multifunctional AChEIs.

This is an initial study to identify simple and efficient core for further design of multi-target-directed ligands (MTDLs) for the treatment of AD. Acetylcholinesterase inhibition

is still one of the most successful therapeutic strategies. The core we disclosed in this paper provides a good starting point. Further studies will be focused on two areas:

1. Improve the inhibitory potency of the compound by occupying the whole binding groove of AChE, including the CAS and PAS site. Structure-guided molecular design will be performed, proper linkers and fragments will be screened and merged into the 4-aminoquinoline core.
2. Considering that many studies report that the progress of AD is tightly correlated to inflammatory condition of nervous system, the design of MTDLs will try to recover the inflammatory environment to normal condition. Antioxidative and component is preferred to be introduced into the core.

### Abbreviations

| | |
|---|---|
| **AChE** | Acetylcholinesterase |
| **AD** | Alzheimer's disease |
| **A$\beta$** | amyloid-$\beta$ |
| **AChEI** | Acetylcholinesterase inhibitor |
| **NMDAR** | N-methyl-D-aspartate receptor |
| **MTDLs** | Multi-target-directed ligands |
| **CAS** | Catalytic site |
| **PAS** | Peripheral site |
| **LE** | Ligand efficiency |

## ACKNOWLEDGEMENTS

We gratefully thank the support of National Natural Science Foundation of China, the Natural Science Foundation of Jiangsu Province, Top-notch Academic Programs Project of Jiangsu Higher Education Institutions, and Priority Academic Program Development of Jiangsu Higher Education Institutions (PAPD).

### Funding

The study was supported from grants 81402851 and 81573281 of the National Natural Science Foundation of China and BK20140957 of Natural Science Foundation of Jiangsu Province. We also received support from the Top-notch Academic Programs Project of Jiangsu Higher Education Institutions (TAPP-PPZY2015A070) and the Priority Academic Program Development of Jiangsu Higher Education Institutions (PAPD). The funders had no role in study design, data collection and analysis, decision to publish, or preparation of the manuscript.

### Grant Disclosures

The following grant information was disclosed by the authors:
National Natural Science Foundation of China: 81402851, 81573281.
Natural Science Foundation of Jiangsu Province: BK20140957.

Top-notch Academic Programs Project of Jiangsu Higher Education Institutions: TAPP-PPZY2015A070.
Priority Academic Program Development of Jiangsu Higher Education Institutions (PAPD).

## Competing Interests
The authors declare there are no competing interests.

## Author Contributions
- Yao Chen, Yaoyao Bian, Sheng Yu, Tingming Fu and Wei Li conceived and designed the experiments, performed the experiments, analyzed the data, contributed reagents/materials/analysis tools, wrote the paper, prepared figures and/or tables, reviewed drafts of the paper.
- Yuan Sun and Chen Kang conceived and designed the experiments, analyzed the data, prepared figures and/or tables, reviewed drafts of the paper, proof reading.
- Yuqiong Pei and Haopeng Sun conceived and designed the experiments, performed the experiments, analyzed the data, contributed reagents/materials/analysis tools, wrote the paper, prepared figures and/or tables, reviewed drafts of the paper, proof reading.

## Data Availability
The raw data has been supplied as Supplemental Dataset.

## Supplemental Information
Supplemental information for this article can be found online at http://dx.doi.org/10.7717/peerj.2140#supplemental-information.

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
