# Peer review of "Identification of 4-aminoquinoline core for the design of new cholinesterase inhibitors"

_PeerJ, doi:10.7717/peerj.2140_

## Round 0.1 · original submission · Minor Revisions

As you see from the reports, both reviewers are supportive. However, both reviewers have some issues with details of the results and in the discussion that you should address by rephrasing these paragraphs. In addition reviewer 1 comments that "The claims made of the docking analysis are somewhat overblown". Make sure that this concern is addressed appropriately. Please also add a reference for CDOCKER .

·

Basic reporting

This article reports the results of an exercise in medicinal chemistry, whereby a known inhibitor was decomposed to discover its active core, which was then derivatised and the subsequent novel molecules tested in an AChE enzyme inhibition assay. The article also describes a docking analysis of the active compounds in an attempt to predict their binding modes. It is reasonably well written although there are quite a few spelling and grammatical errors which must be corrected.

Experimental design

The experiments appear to be have been performed competently, and are on the whole described in sufficient detail. The study is quite "light" in that other than an inhibition assay, no attempt has been made to confirm or validate that the hits are genuine inhibitors and not e.g. coilloidal aggregators. But this fact does not detract from the data, such as it is.

A comment should be added about how the ligand molecules were paramaterised for CHARMm, since this is not trivial. Also, is there a peer-reviewed description of CDOCKER that could be referenced?

Validity of the findings

The claims made of the docking analysis are somewhat overblown. A comparison of two compounds is made in terms of their predicted binding affinity, but no mention is made of how reliable CDOCKER's dG predictions are, or of the error involved. If CDOCKER has an error of +/- 2.5 kcal/mol (as e.g. Autodock does), this means that for compounds 07 and 23 it cannot be claimed that "the binding affinity calculated from docking also supported the phenomenon [of decreased activity]". Also claims are made of pi-pi interactions - does the CHARMm force field even model these?

Table 1 reports purities but does not say how these were determined. Table 2 has Tacrine on the final line, but its potency is reported as an IC50 and so is not comparable to all of the other compounds, which are reported in % inhibition (this should be made clear in the column heading). Why was Tacrine not included as a control in the first place? Since IC50 is the inhibitor concentration at which half the enzyme's activity is inhibited, a conversion to % inhibition should be possible via calculation, and this should be reported alongside.

Why was compound 07 selected for further studies and not 23? Especially since the IC50 of 23 was slightly better.

It is a shame that some computational predictions regarding the compound's ADMET properties were not performed, since these would have been trivial to do.

Suggestions for how these compounds might be progressed would be welcome at the end of the Discussion section. What further analyses and experiments should be performed? Because there are many ...

Reviewer 2 ·

Basic reporting

The standard of English is generally good and the layout of the paper adheres to PeerJ recommendations.
The introduction sets the research in context but lacks some details that are important in understanding the study. Line 38 introduces the concept of a multi-target-directed ligand. For the general reader it is slightly unclear whether this refers to an inhibitor that binds two drug targets or whether it refers to a ligand that hits two sites within an enzyme. It would be clearer if the CAS and PAS sites of AChE were described in a little more detail in the introduction.
Torpedo Californica AChE does not have a particularly high sequence identity to human AChE (52%). I assume the active sites are similar. It might be worth making some comment.
The Figures suffer from lack of structural representations and include typos that make the work difficult to understand and critique.
• Table 1 is not particularly useful in the body of the paper and would be better in supplemental. Serial numbers for supplier data bases are frequently changed. Structural information or IUPAC names stand the test of time.
• Table 2 would be clearer if chemical structures were included rather than names. The numbering system of quinoline would be a useful structural annotation for the non-chemist.
• The experimental data generated is the % inhibition. The term “inhibition rate” not a common term in the literature.
• Including the IC50 of Tacrine in the % inhibition column is confusing.
• Figure 1 - replace inhibitory rate with % inhibition. There is a spelling mistake for concentration. From my reading of the experimental section, the ligand was diluted in a series from 100 microM, hence, the x-axis should read Concentration Log [microM] not Log[M].
• Figure 2 - the x-axis also appears to have a typo and should read microM-1 not M-1 as the substrate was varied between 25 and 425 microM. The inhibitors are quoted as a series of nanoM concentrations. Is this correct? As the concentrations employed seem very small relative to the IC50.
• Figure 4 - The carbon atoms for one ligand are in red. Red is usually reserved for oxygen atoms and is confusing for many readers when next to green.

Experimental design

The study is a useful extension to the author’s first paper. The experimental details are clearly described.
The IC50 data looks clean and technically sound, when not considering the notation errors detailed above.
The Lineweaver-Burke plot does seem to support mixed inhibition but is not very compelling considering the inherent problems in error propagation with this type of plot. My preference would be for a non-linear regression fitting and Km and Vmax reported in a table with errors.

Validity of the findings

The structure-activity observations seem to be supported by the experimental data. The binding modes seem to be a reasonable hypothesis.

---

## Round 0.2 · accepted · Accept

You have addressed the main concerns of the reviewers appropriately. Thank you for the quick reply.